# Consumer Characterization of Commercial Gluten-Free Crackers Through Rapid Methods and Its Comparison to Descriptive Panel Data

**DOI:** 10.3390/foods14172972

**Published:** 2025-08-26

**Authors:** Japneet Brar, Rajesh Kumar, Martin J. Talavera

**Affiliations:** Sensory and Consumer Research Center, Department of Food, Nutrition, Dietetics, and Health, Kansas State University, Olathe, KS 66061, USA; japneetbrar@ksu.edu (J.B.); talavera@ksu.edu (M.J.T.)

**Keywords:** consumer, lexicon, gluten-free crackers, sensory, rapid methods

## Abstract

Despite the continued growth of the gluten-free food market, there is a dearth of sensory and consumer knowledge on commercial products. The existing research is mostly limited to hedonic measurements and ingredient effects instead of analytical methods for a better understanding of product characteristics of gluten-free crackers specifically. In this work, a semi-trained consumer panel used projective mapping to choose objectively different plain/original crackers from a pool of sixteen commercial gluten-free cracker varieties. The cracker samples represented a widespread sensory space originating from different key ingredients such as brown rice, white rice, flaxseed, cassava flour, nut flour blend, millet blend, and tapioca/potato starch blend. Based on projective mapping results, the crackers that mostly represented the sensory space were selected for characterization by a modified flash profiling method. The consumer panel developed 74 descriptors: 30 aromas, 28 flavors, 15 texture terms, and a mouthfeel attribute. The samples were monadically rated for intensity on a 4-point scale (0 = none, 1 = low, 2 = medium, and 3 = high). Rice, toasted, salt, grain, burnt, flaxseed, bitter, earthy, nutty, seeds, and grass were the prevalent aromas and flavors. Others were specific to cracker type. Some of these attributes can be traced back to the ingredients list. Results suggest that ingredients used in small portions are defining the flavor properties over the major grains/flour blends. All samples had some degree of crunchiness, crispness, and pasty mouthfeel; rice crackers were particularly firm, hard, and chewy; brown rice crackers were gritty; crackers with tuber starches/flours were more airy, soft, smooth, and flaky. Overall, the samples shared more aroma and flavor notes than texture attributes. In comparison to trained panel results, consumers generated a greater number of terms and were successful in finding subtle differences primarily in texture but had many overlapped flavors. The developed consumer terminology will facilitate the gluten-free industry to tailor communication that better resonates with consumer experiences, needs, and product values.

## 1. Introduction

The global food market is witnessing a substantial paradigm shift towards specialized dietary options, predominantly propelled by escalating consumer awareness regarding food sensitivities and health-oriented lifestyles. Consumers are seeking gluten-free (GF) food products due to increasing prevalence of celiac disease, non-celiac gluten sensitivity, health trends, convenience, accessibility, and a shift toward plant-based and clean-label diets [1,2,3]. The GF food market has expanded significantly over the past two decades, mostly in baked products including bread, cookies, biscuits, crackers, etc. The market expansion has generated a large amount of research and published works in recent years, mainly focused on improving nutritional value [4,5,6,7], chemical composition [8], technology [9,10,11], quality [12], shelf life [13], and hedonics [14,15]. However, very limited work has been dedicated to understanding consumer terminologies used in describing the perception of sensory characteristics of commercial GF products specifically. As the demand for GF products rises, so does the need for standardized language which can be used to define, evaluate, and market these products.

Recent reviews by Kneževic et al. [12] and Hassan et al. [1] concluded that consumers express a desire for new flavors, textures, and overall improved quality in GF products, indicating a market opportunity for manufacturers to innovate. Similar conclusions were reached by de Kock [16] and Alencar [2]. The existing GF products, which are characterized by diverse ingredients, varying nutritional quality, and sensory traits, could be considered as a new category of foods. Commercially available GF products typically utilize a wide range of ingredients (rice, corn, cassava, potato, nuts, etc.), leading to variety in nutrition, texture and flavor [17]. Some examples are bread varieties [18,19,20], lentil enriched crackers [21], biscuits formulated with buckwheat sorghum lentils [15,22], rice and potato flour biscuits [23], sorghum-based cakes [14], spaghetti [24], muffins [25], and pasta [26]. Moreover, the inclusion of texture binders (hydrocolloids), flavor enhancers, and other functional ingredients could generate a widespread sensory experience [3,17]. Despite the availability of many GF options, the majority of consumers remain dissatisfied with the price, accessibility, and taste of these products. Recent studies highlighted the challenges faced by consumers of GF products. Primary areas of concern were taste, texture, aroma, and overall sensory quality [1]; unacceptable aroma and texture [27]; poor texture, aroma, and taste of GF bread [2]; less delicious GF bread among South American millennials [28]; purchase decisions being mainly influenced by sensory characteristics [29]; dissatisfaction with the sensory experience of bread, pasta, and crackers in the United Kingdom market [30]; lower sensory preference for GF bread, pasta, crackers, flour, and breakfast cereals among the populations of Australia and New Zealand [31]; and low sensorial performance of pasta with Italian consumers [32,33].

In addition, limited knowledge about GF products, including taste features, quality, and benefits, greatly affected consumers’ perception [28]. Several studies have pointed out that knowledge gaps and inadequate education about GF products among consumers played a vital role in product consumption. For example, unawareness about nutritional value [34,35] and restricted knowledge of GF products and their labels affected purchase decisions in some cases [36,37,38,39]. Clearly, GF products are failing to delight consumers due to the lack of a desired sensory profile, and also the lack of awareness among consumers, which has restricted their success in the market [1,3,12,40]. Given these findings, it is essential to not only improve the sensory appeal of GF products but to also educate consumers on products and benefits through targeted awareness campaigns using consumer language. This study has aimed to develop a consumer lexicon as well as to draw a comparison to trained descriptive panel terms.

Among all GF products, gluten-free crackers (GFC) have emerged as a popular snack, appreciated not only by individuals with dietary restrictions but also by health-conscious consumers. At present, these products are marketed and described by various terms on packaging, in advertising, and in online consumer reviews. The existing comprised lexicon surrounding the current range of products is heavily influenced by marketing but completely devoid of product-specific sensory attributes. There remains a gap in comprehending the sensory features and range of commercially available market products [3]. The existing research studies on sensory characteristics of GF foods are constrained to lab-developed prototypes, while only a few have attempted to examine GF breads that are on the market [2,17,18,41]. Therefore, this work is the first of its kind and attempts to generate consumer-driven, intrinsic characteristics of GF crackers. The product selection was restricted to GF crackers that are marketed as plain/original with no added flavors other than salt.

The significance of consumer-based sensory characterization has largely increased in the last decade, partially motivated by the early inclusion of consumer input in the new product development process [42,43]. The relevance of sensory terminology becomes extremely valuable when gluten-free products are perceived as inferior in nutrition, quality, and hedonics [1,3,12]. From conventional to novel profiling techniques, several methodologies are available for gathering information about sensory characteristics of products. Consumer-oriented techniques such as projective mapping (PM) and flash profiling (FP) are based on the evaluation of global similarities and differences among samples [44]. These techniques are less time-consuming, cost-effective, and have been used on various food products to understand consumer perception and description of sensory attributes. In PM, participants focus on relative differences between products by forming an overall opinion on holistic similarities and differences. Participants are instructed to place similar products close to each other, while different products should be located further apart. In this study, PM is used as a tool for preliminary scanning to remove GF crackers that have very similar sensory properties.

The modified flash profiling (MFP) technique allows consumers to generate sensory attributes in their own words and then rank products by their intensities. It also requires less training and has been found to be very effective when used with naïve consumers. Several studies were conducted using FP as a standalone method as well as in combination with traditional descriptive analysis (DA) [45,46]. However, results produced by rapid methods using consumers are difficult to reproduce, less repeatable, and fail to capture fine differences. DA using trained panelists is also widely used to obtain data on subtle differences between products. DA provides detailed sensory profiles and is considered superior due to highly reproducibility and repeatability of results [47,48,49]. The use of a specific method or more than one method is strictly a function of the research objectives and the level of risk tolerance regarding the results.

Henceforth, to make GF products more acceptable among users and successful in the market, it is paramount to not only understand consumer approval but also their perception and description of characteristics. The research approach adopted in this work is pivotal in assessing the nuanced and sophisticated language that consumers use to describe, evaluate, and select gluten-free crackers (GFC). This study aimed to (1) investigate characteristics of commercially available plain GFC using rapid methods with consumers, and to (2) compare consumer terminology with a descriptive lexicon generated by a trained panel.

## 2. Materials and Methods

### 2.1. Participants

Eighteen consumers were recruited from the Kansas City suburban area to participate in the study. All selected participants met predefined requirements of study eligibility through an online screener. Participant ages were between 18 and 65 years old; they were current consumers of GFC, purchased and consumed GFC at least once every 3 months, and were from different income backgrounds. They were also asked about their reasons for consuming GFC. Consumers with gluten sensitivity or who live with someone who has gluten sensitivity, purchase and consume GFC, and do not like eating products with gluten were selected. Those who had participated in any consumer research in the last 3 months and/or were working for a food company or in media/consumer research were excluded. The study was conducted at the Sensory and Consumer Research Center at Kansas State University, Olathe, KS, USA. Compusense (Compusense, Inc., Guelph, ON, Canada) was used for screening, recruitment, execution, and data collection. Participants signed an electronic informed consent form and were compensated for their time. The study was managed under existing Institutional Review Board (IRB) approval (05930) using approved protocols. The number of assessors that participated in this study was as per the recommended range of 10–30 panelists for rapid profiling methods with untrained consumers, subject to sample complexity and panelists’ training [44,50,51,52]. Before each evaluation, an adequate number of training sessions were organized for assessors to familiarize themselves with the methods and products.

### 2.2. Gluten-Free Crackers

Sixteen commercial plain/original flavor GFC were purchased online from Amazon, Walmart, Whole Foods, and Sprouts (Table 1). All cracker samples were inspected for integrity (whole crackers) and were stored in airtight, food-grade 2.8 L containers (Chef’s Path) at ambient conditions until the time of the study. The crackers were from various brands and were made with different grain sources. Crackers with main ingredients such as brown rice, white rice, flaxseed flour, cassava flour blend, nut flour blend, millet blend, and tapioca/potato starch blend were included to ensure variety of flavor and texture.

### 2.3. Projective Mapping

The aim of the PM was to only retain samples that maximize differentiation in the sensory space [44,53,54]. The PM was performed in two sessions. In the first session, the method was explained to the participants, followed by a training session on the selected samples. The assessors were instructed to place the samples on a two-dimensional rectangular sheet based on perceived similarities (similar samples closer and different samples far away from each other). Each consumer was instructed to use his/her own criteria for holistic (appearance, flavor, and texture) differences and similarities. Samples were presented simultaneously; consumers were free to taste the samples in any order and to try them as many times as they wanted. The samples were served at room temperature in 4 oz cups with clear lids (Dart, Mason, MI, USA); the crackers were placed in the cups right before the evaluation sessions to minimize texture changes. Sample cups were labelled with three-digit random codes. Purified bottled water (Niagara Bottling LLC, Diamond Bar, CA, USA) was provided for palate cleansing between samples. After placing the samples, assessors were asked to write down a minimum of three terms that described each sample. Using the PM results, five samples that either did not fit the study scope or had sensory profiles similar to an existing sample in the study were removed. The final ten samples selected for further profiling are highlighted with an asterisk (*) symbol in Table 1. Past studies have used projective mapping for screening of snacks [46], cheese [53], alcoholic beverages [55], strawberry varieties [56], etc.

### 2.4. Modified Flash Profiling

All consumers participated in the MFP task, held in two 90 min sessions over two consecutive days. On day 1, assessors were exposed to the method and products. They were asked to generously generate descriptive terms for the products being evaluated and were asked to avoid hedonic terms such as good, bad, and fair. The samples were served one by one in a monadic order. The evaluation was focused on only 3 modalities in a fixed order: aroma, flavor, and texture. Consumers were instructed to produce descriptive terms and limit the number of terms to 4 for aroma, 4 for flavor, and 3 for texture through individual evaluations. It was emphasized that consumers should focus on the differences they perceived and record attributes following the sequence of perception. The data were collected manually. At the end of day 1, a detailed list of attributes was compiled through a consensus discussion, after removing a few terms which made little sense to the study objectives—for example, bland and no significant taste/aroma. The term fiber was closely related to consumer understanding of grain/seeds, and because those attributes were already present, fiber was eliminated from the final list. The attribute pasty was moved from texture to mouthfeel, as consumers described it as “product dissolving in their mouth”. On day 2, participants were presented with a compiled ballot of 74 attributes collected from the day 1 work. All samples were presented simultaneously for attribute intensity rating on a 4-point scale, with 0 = none, 1 = low, 2 = medium, and 3 = high. The cracker samples were presented in 4 oz cups with clear lids (Dart, Mason, Michigan, USA) labelled with three-digit random codes, and purified bottled water (Niagara) was provided for palate cleansing. Several studies have applied flash profiling to characterize various food products, such as wine [52], fermented soybean curd [57], milk and yogurt products [58], and cheese [59].

### 2.5. Descriptive Analysis

Five highly trained panelists from Sensation Research, Mason, Ohio, evaluated the products using a consensus spectrum method [60]. The panelists had descriptive sensory experience work durations of 7–12 years on various food and beverage product categories including crackers, snacks, meat, beverages, vegetables, meals, etc. Each panelist had conducted more than 1000 h of sensory evaluation on various product categories. The study followed Society of Sensory Professionals (SSP) recommendations for the number of panelists. Past studies have reported 4–18 panelists, but the appropriate number of panelists can vary depending on study type, level of panelist training, previous experience, and product complexity [61,62,63]. The samples were served at room temperature on 4-inch white plates; crackers were taken out from their packages just 5 min before the panel session. A 150-point scale with 1.0-point increments was used for intensity quantification of attributes. The panel evaluated all the samples over three 90-min evaluation sessions, evaluating 4 samples on each of day 1 and day 2, and the remaining 2 samples on day 3. Water was used as the only palate cleanser.

### 2.6. Data Analysis

The XLSTAT version 2025.1.1.1429 (Lumivero, Denver, CO, USA) plugged in Microsoft excel (Microsoft Corporation, Redmond, WA, USA) version 2507 was used to perform data analysis. Multiple factor analysis (MFA) was used for examining the PM data. The coordinates of each sample were measured in inches using a ruler to determine their distance from the X and Y axes. The MFA analysis generated a plot to determine the relationship between the samples. Generalized Procrustes analysis (GPA) was applied to the data collected by the flash profiling method [49,52]. All attributes’ data were run together for deeper understanding through GPA. Attributes that were not elicited or related to the samples were marked as zero for analysis purposes. Principal component analysis (PCA) was applied to the consensus scores of the 44 descriptive attributes produced in DA [64,65]. In sensory studies, it is a standard practice and well documented in the literature to use PCA to represent the relationships among products and attributes obtained using consensus descriptive methods. The RV coefficient [66] was computed to measure the degree of correlation between the MFP and DA methods using XLSTAT. RV coefficient values are between 0 and 1, with values closer to 1 indicating higher similarity.

## 3. Results

### 3.1. Projective Mapping

The PM technique was applied to the selected crackers using a consumer panel. The data obtained in PM were plotted using MFA (Figure 1). Both dimensions (Dim) explained 61.36% of the variability. Flaxseed crackers (DRGF and FAGF) contributed most to both Dims, demonstrating noticeable large sensory differences from the rest of the cracker samples. Crackers with nut-blend-based flour (HUGF and SIMIGF) and brown rice crackers (MAGOGF and MASSGF) influenced the x-axis (Dim 1) to a greater extent, implying that consumers perceived them very differently. For Dim 2, the main contributors were brown rice crackers (SESGF and CRMSFG) and white rice crackers (TJGF and BLDIGF). The distinct text colors of the product names represent different grain sources: orange for brown rice, black for white rice, blue for flaxseed, green for cassava flour, yellow for nut flour blend, red for millet blend, and purple for tapioca/potato starch blend. The spatial placement of crackers on the MFA plot suggests four different product groups. The largest group had eight crackers placed together: GLUTGF, LANCGF, CRUNGF, ABOSGF, SIMIGF, KAMGF, SCHAGF, and HUGF. The crackers in this group had white rice, cassava flour, tapioca/potato starch blend, millet blend, and nut blend as major ingredients in their formulations, respectively. The second largest group had four products: brown rice crackers (SESGF and CRMSGF) and white rice crackers (TJGF and BLDIGF). Additionally, two brown rice crackers (MAGOGF and MASSGF) seem to have very similar sensory characteristics between them, but they were different enough from the other brown rice crackers (SESGF and CRMSFG) to remain more distant. The flaxseed crackers (DRGF and FAGF) were close to each other due to their distinct strong aroma and flavor of flaxseeds. The PM technique was effective; consumers were able to segregate samples based on sensory differences mostly between crackers made with different base flour types. Finally, based on the PM results, only 10 GFC samples (with asterisk (*) symbols in Table 1) were selected for further investigation using MFP and DA.

### 3.2. Modified Flash Profiling

A total of 74 descriptors were generated by consumers after evaluating 10 different plain GFC items. The terms were categorized into 30 aromas, 28 flavors, 15 textures, and one mouthfeel attribute (Table 2). A GPA chart of the MFP is shown in Figure 2. Overall, 41.62% of the variability was explained by the first two Dims. The brown rice samples (MAGOGF and SESGF) are positioned on the positive axis of Dim 1. The main aromas and flavors associated were peanut, flaxseed, nutty, sesame, green, earthy, rice, grain, and seeds. The terms flaxseed and sesame seeds can be associated with ingredients mentioned on product packages. Consumers described brown rice crackers as crunchy, crispy, hard, gritty, and with subdued levels of moisture. One white rice cracker (BLDIGF) had sensory features similar to those of brown rice crackers, with main characteristics of corn, rice, grain, and bitter. The other white rice samples (GLUTGF and LANCGF), the cassava flour sample (CRUNGF), and the tapioca/potato starch sample (ABSOGF) were positioned on the negative axis of Dim 1. These were mainly characterized by toasted, flour, salt, smoothness, and puffiness. The tuber-flour- and starch-formulated crackers were particularly associated with uncooked flour, burnt, earthy, airy, and firmness. Consumers identified specific flavors such as butter, chemical, oil, thickness, and puffiness in one white rice sample (LANCF). This sample was also noted for oxidized oil, probably originating from palm oil being the leading ingredient in the list.

The second dimension explained 14.64% of the data; the nut blend cracker (SIMIGF) and one white rice cracker (KAMGF) are positioned closer to each other. The SIMIGF sample was distinct for herbs, chicken, savory, cheesy, onion, and butter flavors. The millet blend sample SCHAGF stretched the plot with its distinct sensory properties. It was related with toasted, cardboard, pepper, wheat, sweet, oats, flakiness, roughness, and pasty mouthfeel. The brown rice crackers (SESGF and MAGOGF) were noticeably different in texture features, mainly crispiness, grittiness, hardness, and crunchiness. Consumers associated rice crackers with the highest hardness and crispiness [67]. Two white rice crackers (LANCGF and BLDIGF) were characterized by thickness and puffiness, while GLUTGF was gummy and chewy, and the cassava flour (CRUNGF) crackers were distinct for their firmness and airy texture. Giuberti et al. [68] also reported increase in hardness with high levels of rice flour in GF cookies. The tapioca/potato starch blend (ABSOGF) crackers were described more as soft, smooth, and airy. Both the millet blend (SCHAGF) and nut blend (SIMIGF) crackers were characterized by flakiness and pasty mouthfeel. The results suggest that there is no clear relationship between texture attributes and crackers formulated with specific grain types, except for brown rice crackers, which were positioned together in the MFA (Figure 2). Overall, consumers were able to perform the MFP task, and no problems were reported during the study. Consumers narrated the perceived characteristics using their own language, and the words provided to describe stimuli were specific to the modality. It is worth noting that consumers described aroma (sniffed through nostrils) first, and flavor during mastication. There are some attributes perceived both in aroma and flavor evaluation. The results suggest that consumers stayed focused on the task as well as on the specific modality of interest. The cracker samples were well differentiated for texture, but less so in terms of flavor and aroma.

### 3.3. Descriptive Analysis

Trained panelists for the DA generated 44 terms which were classified into 7 appearance, 30 flavor, and 7 texture attributes (Table 2). PC1 (30.71%) and PC2 (21.93%) explained 52.64% of the variance in the DA data (Figure 3). The positive axis of PC1 was characterized by brown rice (MAGOGF and SESGF), white rice (BLDIGF), and nut blend (SIMIGF) crackers, whereas samples of millet blend (SCHAGF), tuber flour (ABSOGF and CRUNGF), and white rice (GLUTGF, LANCGF, and KAMGF) crackers were on the negative axis of PC1.

The brown rice crackers (MAGOGF and SESGF) were clearly different from the other samples regarding the amount and size of seeds, roughness emerging from seeds, and earthy, burnt, and cardboard characteristics. Nevertheless, the nut blend (SIMIGF) crackers and one variety of white rice (BLDIGF) crackers had different grain types but were positioned similarly. They were characterized by brittle texture (fracturability), nutty, and oily flavor. The samples SCHAGF, ABSOGF, GLUTGF, and LANCGF were located on the positive axes of PC1 and PC2, mainly described as thick, bitter, baking soda, toasted, and uneven browning. The millet blend sample was associated with unique characteristics of wheat, sesame/flax seeds, and toasted; the tapioca flour cracker (ABSOGF) was described as dry, potato, and burnt; two white rice crackers (GLUTGF and LANCGF) had starch complex and rice notes. The cassava flour crackers (CRUNGF) and one white rice cracker (KAMGF) were distinct for their shiny appearance but shared many sensory properties similar to other white rice crackers (GLUTGF and LANCGF). The DA results indicated that the GFC samples were relatively well differentiated in appearance, texture, and flavor. In addition, the samples could be largely grouped by their similarities, such as those of white rice crackers (GLUTGF and LANCGF) and brown rice crackers (MAGOGF and SESGF). However, the spectrum of differences is less varied and diverse, probably due to the limited range of GF crackers available in the market [12].

## 4. Discussion

### 4.1. Projective Mapping

At large, the PM results indicate that crackers with the same flour type are not necessarily placed together, and there are certainly common sensory properties dispersed within and between groups. Particularly, consumers did not perceive noticeable differences between pairs of brown rice crackers (SESGF and CRMSFG) and pair (MAGOGF and MASSGF). At this stage, the cracker CRMSGF (brown rice) was eliminated because the similar-profile brown rice cracker SESGF was retained. Following the same strategy, MAGOGF was selected to move forward over MASSGF. Similarly, the white rice cracker TJGF was removed, as it seemed to have a similar profile to BLDIGF. Using the PM results, both flaxseed crackers FAGF and DRGF were not taken forward because of strong flavor originating from flaxseed, which was beyond the scope of this study. The differences were so remarkable that the researchers believed that these products belonged to a different category. Correspondingly, HUGF was eliminated because of strong onion and garlic notes which deviated from the study objective of including plain/original crackers so that the effects of different flour blends could be observed on the different sensory attributes. The 10 samples selected to move forward for MFP are highlighted with an asterisk (*) symbol in Table 1. Overall, the PM technique was valuable to narrow down a large number of samples for further evaluation. Likewise, other studies have also used PM for preliminary examination of snack foods [46], dairy products [69], and other products [54].

### 4.2. Modified Flash Profiling

The results of MFP demonstrated that GFC samples were primarily classified by their aromas and flavor notes of seeds, flour, rice, toasted, salt, grain, earthy, etc. A few unique aromas and flavors were specific to certain cracker types such as onion and cheese (SIMIGF), rosemary (CRUNGF), flaxseed (MAGOGF), and oats (SCHAF). These samples had detectable flavors which could be related to one of the ingredients listed on the package, but they were still marketed as “plain” crackers. The brown rice (MAGOGF and SESGF), millet blend (SCHAG), and nut blend (SIMIGF) crackers were well differentiated from the other samples. However, in case of white rice (KAMGF, GLUTGF, and LANCGF), cassava flour (CRUNGF), and tapioca/potato starch (ABSOGF), the crackers were not as clearly distinguished because of their overlapping flavors (rice, toasted, and flour) and texture characteristics (crunchy, crispy, hardness, and grittiness). The substantial overlap of flavor and aroma terms between samples suggests that consumers have not been able to clearly differentiate aromas and flavors, resulting in some redundancy (Table 2). It is evident, however, that the MFP methodology is a good fit in a scenario where broader characterization is needed instead of detecting subtle differences in specific attributes.

### 4.3. Descriptive Analysis

In general, all samples had some degree of shine, thickness, and rough surface, mainly from inclusions such as flaxseeds. The prevalent flavor attributes across samples were salt, sweet, astringent, rice, starch complex, toasted, baking soda, bitter, sour, oily, cardboard, and strong lingering aftertaste. The common flavor attributes can be linked to base ingredients such as white rice, brown rice, millet blend, and tapioca starch. Some flavor terms such as coconut, seedy, wheat-like, peppery heat, earthy, black pepper, oil, herbs, soy sauce, garlic, seaweed, dairy, and nutty were specific to certain cracker types. For example, coconut flavor for cassava flour (CRUNGF); wheat-like flavor in millet blend (SCHAGF); dairy in white rice (LANCGF); seaweed, soy sauce, and burnt in brown rice (MAGOGF); and peppery heat in brown rice (SESGF). It appears that ingredients such as almonds, non-fat dairy milk, flax seeds, sesame seeds, and coconut flour influenced the differences through rare flavors. Similarly, texture attributes such as thickness, hardness, fracturability, grit, dryness, and tooth packing were present across the cracker samples used in this study. The panel also identified that subtle texture differences in roughness from seeds/particulates occurred only for a few samples. Overall, the trained panel was able to differentiative GFC samples by shared sensory characteristics as well as by specific attributes.

### 4.4. Comparison Between Modified Flash Profiling and Descriptive Analysis

As expected, the MFP results from the consumer panel produced a greater number of attributes in comparison to the trained panel [58]. The consumer panel produced 74 descriptors (30 aromas, 28 flavors, 15 textures, and 1 mouthfeel attribute), whereas the DA panel generated 44 terms (7 appearances, 30 flavors, and 7 textures) to characterize GFC. The consumer panel was instructed not to use appearance for profiling. The trained panel generated 30 terms for aroma/flavor combined, and the consumer panel generated 58 aroma and flavor terms. Table 3 highlights the summary of various indices between the DA and MFP methods. The RV coefficient was 0.62, indicating moderate similarities between the results obtained from the MFP and DA methods.

A comparison of aroma and flavor attributes from both panels reveals 16 common terms (Table 2). The terms are bitter, burnt, pepper/back pepper, cardboard, dairy/buttery, garlic, herbs, oily, salty, sweet, toasted, wheat-like/wheat, sesame, seed, flax, and earthy. The DA panel combined the terms sesame, seedy, and flax into a single descriptor, whereas consumers saw them as three different attributes. The terms that were only provided by the DA panel for describing aromas and flavors of GFC are baking soda, dairy/nut milk (which was described as cheesy/buttery by consumers), soy sauce, seaweed, irritating (which might be ascribed to chemicals by consumers), burning heat from pepper, true to gluten, sour, potato (flour/starch), starch complex, coconut/coconut flour, astringent, overall aroma intensity, overall flavor intensity, and aftertaste. Consumers produced unique flavors such as rosemary, sunflower seed, woody, uncooked flour, green, grassy, oxidized/rancid oil, etc., demonstrating the effectiveness of using category users as panelists.

A comparison between the biplots generated by MFP and DA shows that several GFC samples are positioned very similarly. The brown rice crackers (MAGOGF and SESGF) are positioned in the same sensory space along with one of the white rice cracker samples (BLDIGF). The terms used to describe brown rice crackers such as earthy, flaxseed, and cardboard were common. Likewise, the cracker samples ABSOGF, GLUTGF, and LANCGF are also positioned together. Both panels described SCHAGF and CRUNGF as sweet, LANGF as strong dairy (butter), SIMIGF as onion, CRUNGF as garlic, and SCHAGF and SIMIGF as having herbs and pepper flavor. Consumers associated corn flavor with the BLDIGF and SESGF cracker samples, but the trained panel did not. Interestingly, corn was present in 3 out of 10 formulations. Another term of coconut for aroma/flavor was generated by the DA panel and reported in two formulations; however, this term was not identified by the consumer panel. Additionally, the attribute sunflower seed was used by the consumer panel, which was on the ingredient list for MAGOGF, but it was not part of attributes generated by the DA panel. It can be said that rapid methods can be used as an initial technique to identify key sensory descriptors of products using consumers, mainly for developing marketing and consumer-friendly language. To obtain sensory data which are reproducible, accurate, and more sensitive to small differences, and have standardized terms that are clearly defined, results from a descriptive panel are more appropriate.

## 5. Conclusions

This study helps increase the knowledge about the characteristics of commercially available GFC. Overall, it can be concluded that GFC market products lack sensory variety and complexity. This study suggests that a different base flour does not necessarily produce a diverse sensory experience. However, this hypothesis needs to be further evaluated in a more controlled setting, ideally using a design of experiments (DOE) approach. In addition, minor ingredients such as flaxseed or onions might dominate flavor profiles and also create noticeable and unique flavors across samples, even though they are still marketed or labelled as “plain” crackers. MFP evaluations by a consumer panel used 74 attributes, and the descriptive panel used 44 terms to explain sensory features of GFC. Among both panels, flavor attributes were more common than texture attributes, and the sample grouping and positioning were somewhat identical. While the descriptive panel captured subtle differences between samples, the consumer panel identified unique flavor notes at lower intensities. The findings of this work can serve as a guide for product improvement, product development, quality control, examining the effects of ingredients on product properties, designing marketing campaigns, and understanding consumer experiences.

The findings demonstrate that rapid methods can be viable and produce actionable results when compared to traditional descriptive methods. This study was performed at a central location under controlled settings; future studies can explore consumer perception in more dynamic and situational conditions. Additionally, forthcoming research could focus on evaluating sensory properties of GFC eaten in combination with dips, spreads, toppings, etc.

## Figures and Tables

**Figure 1 foods-14-02972-f001:**
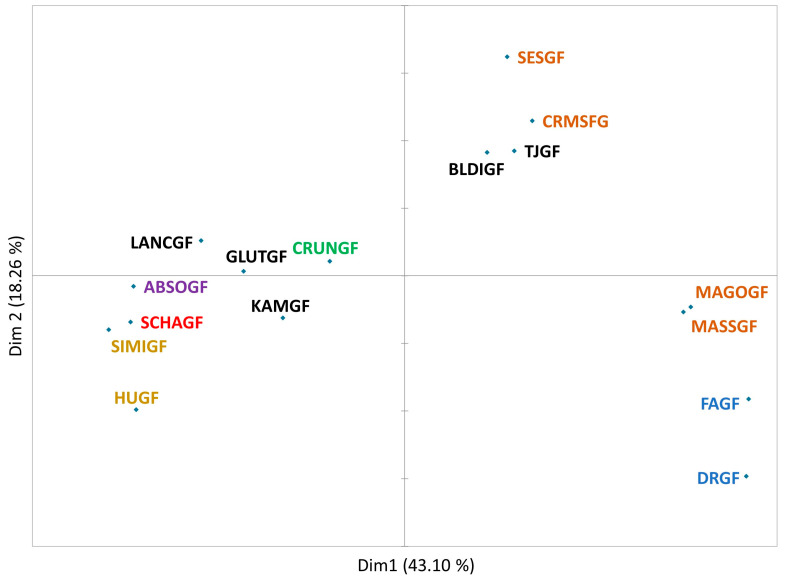
Projective mapping result plot generated by multiple factor analysis for gluten-free crackers. The distinct text colors of product names represent different grain sources: orange for brown rice, black for white rice, blue for flaxseed, green for cassava flour, yellow for nut flour blend, red for millet blend, and purple for tapioca/potato starch blend. Refer to Table 1 for detailed names of cracker varieties.

**Figure 2 foods-14-02972-f002:**
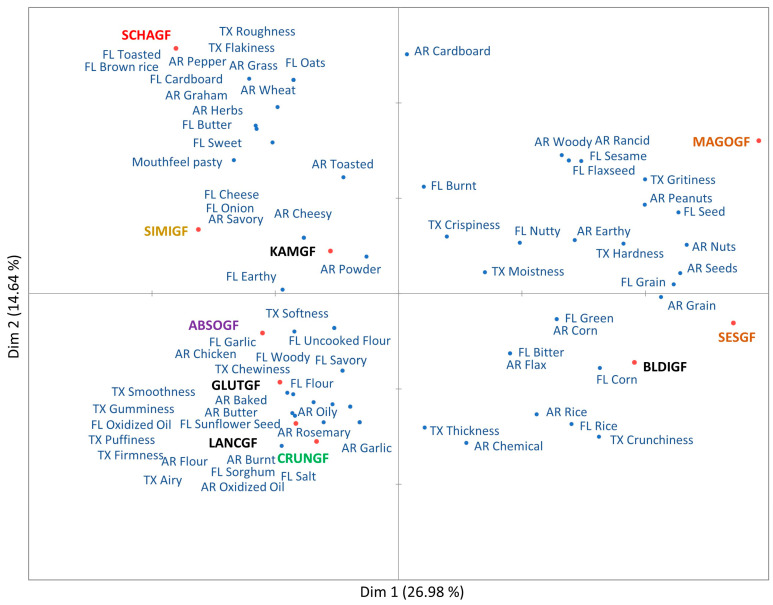
Modified flash profiling result plot generated by generalized Procrustes analysis for gluten-free crackers. The distinct text colors of product names represent different grain sources: orange for brown rice, black for white rice, green for cassava flour, yellow for nut flour blend, red for millet blend, and purple for tapioca/potato starch blend. Attributes are represented in blue-colored text. Refer to Table 1 for detailed names of cracker varieties. Note: AR—aroma, FL—flavor, and TX—texture.

**Figure 3 foods-14-02972-f003:**
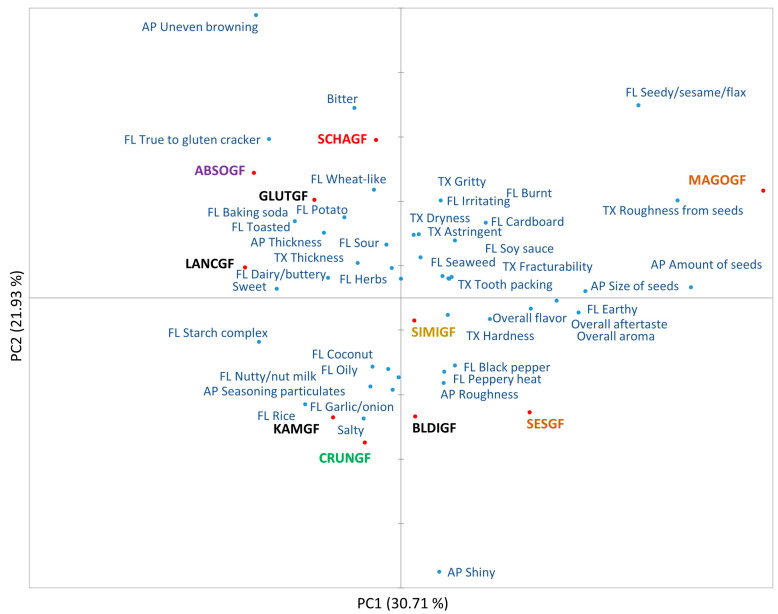
Principal component analysis chart of descriptive data. The distinct text colors of product names represent different grain sources: orange for brown rice, black for white rice, green for cassava flour, yellow for nut flour blend, red for millet blend, and purple for tapioca/potato starch blend. Attributes are represented in blue-colored text. Refer to Table 1 for detailed names of cracker varieties. Note: AP—appearance, FL—flavor, and TX—texture.

**Table 1 foods-14-02972-t001:** List of commercially available gluten-free crackers used for consumer and descriptive lexicon development.

Cracker Brand	Code	Flour Base	Ingredients	Variety
Absolutely Gluten-Free Crackers *	ABSOGF	Tapioca/potato starch blend	Tapioca starch, water, potato starch, potato flakes, palm oil, honey, egg yolks, natural vinegar, salt	Original
Crunchmaster Grain-Free Crackers *	CRUNGF	Cassava flour	Cassava flour, organic coconut flour, tapioca starch, safflower oil, sea salt, garlic powder	Original
Hu Gluten-Free Grain-Free Crackers	HUGF	Almond, cassava, coconut flour blend	Grain-free flour blend (almond, cassava, organic coconut), black chia seed, flax seed, organic coconut aminos	Sea Salt
Schar Table Gluten-Free Crackers *	SCHAGF	Millet blend	Non-GMO corn starch, vegetable fats and oils (palm, palm kernel, non-GMO rape seed), maltodextrin, modified tapioca starch, whole millet flour, non-GMO soy flour, rice syrup, whole rice flour, buckwheat flour, sorghum flour, flax seed flour, non-GMO corn flour, dried sourdough (buckwheat, quinoa), non-GMO soy bran, poppy seeds, non-GMO sugar beet syrup, sea salt, cream of tartar, ammonium bicarbonate, baking powder, guar gum, modified cellulose, citric acid, natural flavoring (rosemary)	Original
Simple Mills Sea Salt Crackers *	SIMIGF	Nut flour blend	Nut and seed flour blend (almond flour, sunflower seeds, flax seeds), tapioca starch, cassava, organic sunflower oil, sea salt, organic onion, organic garlic, rosemary extract (for freshness)	Sea Salt
Glutino Gluten-Free Crackers *	GLUTGF	White rice	Corn starch, white rice flour, organic palm oil, modified corn starch, eggs, sugar, salt, vegetable fibers, dextrose, guar gum, sodium bicarbonate, natural flavor, monocalcium phosphate, ammonium bicarbonate	Original
Blue Diamond Nut Thins *	BLDIGF	White rice	Rice flour, almonds, potato starch, sea salt, safflower oil, natural flavors (contains milk)	Original
Lance Gluten-Free Crackers *	LANCGF	White rice	Palm oil, rice flour, rice starch, sugar, corn starch, potato starch, baking soda, tapioca flour, glucose, xanthan gum, monocalcium phosphate, salt, soy lecithin, locust bean gum, non-fat milk	Original
Trader Joe’s Savory Thin Crackers	TJGF	White rice	Rice, sesame seeds, expeller-pressed safflower oil, tamari soy sauce (soybeans, rice, salt), salt, garlic, soybean	Original
Mary’s Gone Crackers *	MAGOGF	Brown rice	Brown rice, quinoa, flax seeds, sesame seeds, tamari (water, soybeans, salt, vinegar), sea salt	Original
Sesmark Gluten-Free Crackers *	SESGF	Brown rice	Rice flour, expeller-pressed safflower oil, sesame seeds, sesame flour, wheat-free tamari soy sauce powder [tamari soy sauce (soybeans, salt), maltodextrin (from corn)], wheat-free teriyaki powder, [wheat-free teriyaki sauce (tamari soy sauce ([soybeans, salt]), sake (rice, salt), apple cider vinegar, garlic, mustard, ginger, white and black pepper), maltodextrin, sucrose, fructose], onion powder, soy lecithin	Sea Salt
Mary’s Gone Super Seed Gluten-Free Crackers	MASSGF	Brown rice	Brown rice, quinoa, pumpkin seeds, sunflower seeds, flax seeds, sesame seeds, poppy seeds, sea salt, seaweed, black pepper, spices	Original
Crunchmaster Multigrain Crackers	CRMSGF	Brown rice	Brown rice flour, whole grain yellow corn, potato starch, safflower oil, oat fiber, cane sugar, sesame seeds, flax seeds, millet, sea salt, quinoa seeds.	Original
Ka Me Rice Crackers *	KAMGF	Jasmine rice	Jasmine rice, rice bran oil, sea salt, soybean tocopherols (preservative)	Original
Doctor in the Kitchen Flackers	DRGF	Flaxseed	Organic flax seeds, organic apple cider vinegar, sea salt	Sea Salt
Foods Alive Original Flax Crackers	FAGF	Flaxseed	Golden flaxseed, Bragg Liquid Aminos (a non-GMO wheat-free soy sauce), lemon juice	Original

Note: Gluten-free crackers with an asterisk (*) symbol were selected for modified flash profiling and descriptive analysis.

**Table 2 foods-14-02972-t002:** List of attributes generated by consumers using modified flash profiling method and by trained panel with descriptive analysis.

Appearance		Aroma and flavor			Texture	
Modified Flash Profiling	Descriptive	Modified Flash Profiling		Descriptive	Modified Flash Profiling	Descriptive
		Aroma	Flavor	Aroma and Flavor		
None	Amt of seeds/inclusions	Baked	Bitter	Astringent	Airy	Dryness/moisture absorbency *
	Color	Burnt	Brown rice	Baking soda	Chewiness	Fracturability
	Holes (yes/no)	Butter	Burnt	Bitter *	Crispiness	Grit/chalky/mouth coating *
	Rough appearance	Cardboard	Butter	Black pepper *	Crunchiness	Hardness *
	Seasoning particulates	Cheesy	Cardboard	Burning heat from pepper	Firmness	Roughness(seeds/particulates) *
	Shape	Chemical	Cheese	Burnt *	Flakiness	Thickness *
	Shiny	Chicken	Corn	Cardboard *	Grittiness	Tooth stick/tooth packing
	Size of seeds	Corn	Earthy	Coconut (flour)	Gumminess	
	Thickness appearance	Earthy	Flaxseed	Dairy/buttery *	Hardness	
	Uneven browning	Flaxseed	Flour	Earthy *	Moistness	
		Flour	Garlic	Garlic/onion *	Puffiness	
		Garlic	Grain	Herbs *	Roughness	
		Graham	Green	Irritating	Smoothness	
		Grain	Nutty	Nutty/nut milk *	Softness	
		Grass	Oats	Oily *	Thickness	
		Herbs	Onion	Overall aftertaste		
		Nuts	Oxidized oil	Overall aroma		
		Oily	Rice	Overall flavor		
		Oxidized oil	Salt	Potato (flour, starch)		
		Peanuts	Savory	Rice (flour, starch) *		
		Pepper	Seed	Salty *		
		Powder	Sesame	Seaweed		
		Rancid	Sorghum	Seedy/sesame/flax *		
		Rice	Sunflower seed	Sour		
		Rosemary	Sweet	Soy sauce		
		Savory	Toasted	Starch complex		
		Seeds	Uncooked flour	Sweet *		
		Toasted	Woody	Toasted *		
		Wheat		True to gluten cracker		
		Woody		Wheat-like *		

Note: Descriptive attributes with asterisk (*) symbols were also found in consumer terms obtained in modified flash profiling.

**Table 3 foods-14-02972-t003:** Summary of modified flash profiling method and descriptive analysis.

	Modified Flash Profiling	Descriptive Analysis
Number of products evaluated	10	10
Number of panelists	18	5
Number of sessions	3	3
Panel type	Untrained (individual evaluations)	Trained (consensus)
Task	Used own words to describe the attributes. Rated products’ perceived attributes for intensities	Rate products for intensities. Panelists were trained on specific attributes and references
Scale used	4-point scale (0 = none, 1 = low, 2 = medium, and 3 = high)	150-point scale with 1.0-point increments
Data analysis type	Multiple factor analysis	Principal component analysis
Total number of attributes	74	44
Appearance	-	7
Aroma	30	-
Flavor	28	30
Texture	15	7
Mouthfeel	1	-
Results	Global overview of commercial gluten-free crackers space. Ideal for obtaining consumer differentiation.	Precise, accurate, and consistent measurements between cracker samples.

## Data Availability

The original contributions presented in the study are included in the article. Further inquiries can be directed to the corresponding author.

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
