# Peer review of "Consumer Characterization of Commercial Gluten-Free Crackers Through Rapid Methods and Its Comparison to Descriptive Panel Data"

_foods, 2025, doi:10.3390/foods14172972_

Round 1

Reviewer 1 Report

Comments and Suggestions for Authors

The topic of manuscript is relevant in both the food science and in the food industry as the consumers' opinion about the sensorial properties of food products is a key factor in marketing.

The manuscript is well written and clearly organized. The introduction is rather long, I would recommend to shorten and simplify that, focusing on the sensorial methods.

Description of experiments is clear, detaled and easy to follow.

Tables and Figures are well constructed.

However, I think, major revision of the research project is needed to provide useful information. The main concerns about the results presented are the followings:

  • too few panelists are involved in the study. Therefore all the results seem to be too subjective. Especially because panelists were not trained experts.
  • too many food products were analyzed at the same time
  • wide variation in main ingredient of crackers (ie rice, nuts etc.) determine that there will be big differences in the properties investigated. Therefore results will be not very well applicable when more similar products would be compared.
  • the above mentioned concerns are quite well seen in Figure 2, in which Dim1 describes 26.98% while Dim2 14.64% of the variation, which seems to be rather low.
  • conclusions are too general

Author Response

Dear reviewer

Thank you very much for reviewing our manuscript and providing us with valuable comments. We have made changes to the article in accordance with your suggestions. The changes in the manuscript are highlighted in yellow color.

Please see our specific responses below:

COMMENTS FROM REVIEWER 1

Comment 1: The topic of manuscript is relevant in both the food science and in the food industry as the consumers' opinion about the sensorial properties of food products is a key factor in marketing.

Author’s response: The authors appreciate reviewer's feedback regarding the relevance of our research work in the manuscript.

Comment 2: The manuscript is well written and clearly organized. The introduction is rather long, I would recommend to shorten and simplify that, focusing on the sensorial methods.

Author’s response: Thank you for the feedback. The specific goal of the introduction section was to highlight the existing knowledge gap in the consumer perception of sensory characteristics of gluten free products, including crackers. The primary focus of this research was to generate sensory terminology using consumers through rapid methods, explaining the sensory characteristics, and comparing these results with a trained panel as a benchmark, but not necessarily comparing the two methods. There are many studies that have published results comparing effectiveness and consistency between rapid and traditional descriptive analysis. The authors discussed the methodology used in this study between Lines 101 and 124.

Comment 3: Description of experiments is clear, detailed and easy to follow.

Author’s response:  Thank you for the positive comments.

Comment 4: Tables and Figures are well constructed.

Author’s response: Thank you for the positive comments.

Comment 5: However, I think, major revision of the research project is needed to provide useful information. The main concerns about the results presented are the followings:

Comment 6: too few panelists are involved in the study. Therefore all the results seem to be too subjective. Especially because panelists were not trained experts.

Author’s response: The authors respect the reviewer’s perspective. This study used 18 assessors that are within the recommended range of 10-20 panelist for rapid profiling methods with untrained panelists. No hedonic data is being collected. Rapid profiling aims at using consumers in a more analytical manner, therefore lower samples sizes are expected. Additionally, two training sessions were  organized for participants to get acquainted with method and products. The authors believe that the number of assessors used were in-line with existing  recommendations and published studies. The sources were cited in the manuscript see Lines 148-152

Comment 7: too many food products were analyzed at the same time

Author’s response: In an analytical setting (non-hedonic), the advised number of products to evaluate can vary from 4 to 20, depending on the complexity of the product and fatigue level of panelists. This study evaluated plain gluten free crackers which are not considered complex product. The panel was given appropriate time and breaks; and palate cleansers were used to minimize palate saturation and fatigue. In addition, it is a common practice for trained panels to objectively evaluate a larger number of samples over several sessions vs. hedonic testing.

Comment 8: wide variation in main ingredient of crackers (ie rice, nuts etc.) determine that there will be big differences in the properties investigated. Therefore, results will be not very well applicable when more similar products would be compared.

Author’s response: The review comment summarized the hypothesis tested in this work. A wide range of crackers from different brands, formulated with various base flours were selected to include diversity in the sensory space, which, actually, has the goal of maximizing its applicability. In any case, results indicated that commercial gluten free crackers lacked sensory differences and complexity.  The differences were mainly due to minor ingredients instead of base flours.

Comment 9: the above mentioned concerns are quite well seen in Figure 2, in which Dim1 describes 26.98% while Dim2 14.64% of the variation, which seems to be rather low.

Author’s response: Both dimensions explained a cumulative variation of 41.62%. The authors agree, this overall amount is on the low side and could represent some loss of information. However, given that the main goal is to show the terms generated and some of their general relationships as an example, we feel that adding more graphs is beyond the scope and will unnecessarily increase the length of the manuscript.

Comment 10: conclusions are too general

Author’s response: The authors respectfully disagree with the reviewers’ comments. We feel that the conclusion section does include very specific findings between Lines 449-460, which are in line with the general findings of the research. The findings include 1) “GFC market products lack sensory variety and complexity”, 2) “a different base flour does not necessary produce a diverse sensory experience”, 3) “minor ingredients such as flaxseed or onions might dominate flavor profiles and also create noticeable and unique flavors across samples, even though they are still marketed or labeled as “plain” crackers”, 4) MFP evaluations by a consumer panel used 74 attributes, while the descriptive panel used 44 terms to explain sensory features of GFC”, 5) Among both panels, flavor attributes were more common than texture”, 6) “the sample grouping and positioning were somewhat identical between panels”, and 7) “while the descriptive panel captured subtle differences between samples, the consumer panel identified unique flavor notes at lower intensities”. Lines 452- 462.

Reviewer 2 Report

Comments and Suggestions for Authors

Article “Consumer characterization of commercial gluten-free crackers through rapid methods and its comparison to descriptive panel”  used 3 different methodologies to describe GFC and compare them. The article is not quite new regarding the approach as a whole, but it bring interesting information for the GF area.

In the abstract the sentence “The existing research is mostly limited to hedonic measurements and ingredient effects instead of analytical methods for a better understanding of product characteristics” is not true. There are several works currently dealing with this issue.

Introduction “Though, very limited work has been dedicated to understanding consumer terminologies used in describing the perception of sensory characteristics of GF products” Again, I am not realy sure about this sentence.

Methodology

PM section: please, add references.

Data analysis section: “Principal Component Analysis (PCA) was applied to the consensus  scores of the 44 descriptive attributes produced in DA”. This is not proper method to evaluate consensus scores. Authors should evaluate the interaction between panelist*sample to be not significant to check the goodness of the DA results.

Results

PCA is an exploratory analysis and I believe ANOVA is critical to evaluate differences among samples.

Also, it is important to evaluate the RV parameter to check the  relation among the methodologies used in the present work.

After this changes, I believe it would be ok to evaluate the discussion section and the conclusions

Author Response

Dear reviewer

Thank you very much for reviewing our manuscript and providing us with valuable comments. We have made changes to the article in accordance with your suggestions. The changes in the manuscript are highlighted in yellow color.

Please see our specific responses below:

COMMENTS FROM REVIEWER 2

Comment 1: Article “Consumer characterization of commercial gluten-free crackers through rapid methods and its comparison to descriptive panel”  used 3 different methodologies to describe GFC and compare them. The article is not quite new regarding the approach as a whole, but it bring interesting information for the GF area.

Author’s response: The authors appreciate the reviewer's feedback regarding the significance of our research work in the manuscript. However, we respectfully disagree with this specific comment. This is the first research work that has produced a consumer and trained panel lexicon for commercial gluten free crackers in the United States market. The authors does highlight this point in the literature review section, mentioning that existing research work is highly focused on lab developed crackers and biscuits and not commercial products. Lines 93-95

Comment 2: In the abstract the sentence “The existing research is mostly limited to hedonic measurements and ingredient effects instead of analytical methods for a better understanding of product characteristics of GF crackers specifically” is not true. There are several works currently dealing with this issue.

Author’s response: The authors are thankful for reviewers’ comment. The statement was mentioned in the context of commercial gluten free crackers. The existing work in the literature is done on lab developed products and focused on measuring consumer liking and quality ratings instead of a generally applicable lexicon. The authors have highlighted exiting research work and its limitations between Lines 93-95.

Comment 3: Introduction “Though, very limited work has been dedicated to understanding consumer terminologies used in describing the perception of sensory characteristics of commercial GF products specifically” Again, I am not realy sure about this sentence.

Author’s response: This comment is similar to the above mentioned comments. The authors have already addressed it previously.

Methodology

Comment 4: PM section: please, add references.

Author’s response: The references are added between Lines 164-165 and Lines 180 to 182.

Comment 5: Data analysis section: “Principal Component Analysis (PCA) was applied to the consensus  scores of the 44 descriptive attributes produced in DA”. This is not proper method to evaluate consensus scores. Authors should evaluate the interaction between panelist*sample to be not significant to check the goodness of the DA results.

Author’s response: The authors respectfully but strongly disagree. Conventional descriptive analysis methods include both consensus and individual data with replications. Both are equally valid and have specific applications. It is a common practice in sensory data to analyze consensus descriptive data with principle component analysis. Most published lexicon development work involves consensus data with PCA exploration. The interaction between panelist*sample can be conducted on individual scores with repetitions. Analysis of Variance cannot be applied on consensus data because there is no variability. This study used a consensus method to generate descriptive profiles of commercial gluten free crackers instead of individual evaluations. References for the advised principle component analysis method are cited in Lines 233-237.

Results

Comment 6: PCA is an exploratory analysis and I believe ANOVA is critical to evaluate differences among samples.

Author’s response: Similar to the above-mentioned comment. This study used consensus method to generate descriptive profiles of commercial gluten free crackers. ANOVA can only be performed on individual scores with repetitions not on consensus descriptive data.

Comment 7: Also, it is important to evaluate the RV parameter to check the  relation among the methodologies used in the present work.

Author’s response:  The authors appreciate the reviewer’s feedback. The RV coefficient is calculated following the recommended methodology in Line 234-237. The findings are explained in Line 414-415.

Comment 8: After this changes, I believe it would be ok to evaluate the discussion section and the conclusions

Round 2

Reviewer 2 Report

Comments and Suggestions for Authors

Dear Editors,

I do not agree PCA is the best way to analyze the data. Authors clearly disagree.

I suggest another reviewer to make the peer review, since this is the main point of this article.

Kind regards

Author Response

Dear reviewer,

Thanks again for reviewing our manuscript and providing us with valuable feedback. The authors have addressed the reviewers’ comments from round 1 in the earlier revised submission and also made several changes and additions based on reviewers’ feedback at that time.

Reviewer’s comment (Reviewer report round 2) 

I do not agree PCA is the best way to analyze the data. Authors clearly disagree.

I suggest another reviewer to make the peer review, since this is the main point of this article.

Authors response: As we mentioned in our prior response to this comment, we respectfully disagree with the reviewer’s comment regarding the use of PCA (principal component analysis). In sensory studies, it is a standard practice to use PCA to represent the relationships among products and attributes obtained using consensus descriptive methods. This approach is well documented in the literature, with numerous published studies employing similar methodologies. Furthermore, the primary focus of our paper is to generate sensory terminology using consumers through rapid methods, describe sensory characteristics, and compare the results with those from a trained panel as a benchmark. Therefore, the authors believe the reviewer’s most recent (round 2) comment on the revised submission does not warrant further changes in the manuscript. The authors have added a statement in the data analysis section to reflect the same understanding. See line 234-236 (highlighted in yellow color).

Here are a few references with recommendations on the use of PCA in sensory research and published articles that have used a very similar approach.

  1. Society of Sensory Professionals. https://www.sensorysociety.org/knowledge/sspwiki/Pages/PCA.aspx
  2. Lawless, H. T., & Heymann, H. (2010). Sensory evaluation of food: principles and practices. Springer Science & Business Media.
  3. Chambers IV, E. (2018). Consensus methods for descriptive analysis. Descriptive analysis in sensory evaluation, 211-236.
  4. Suwonsichon, S., Chambers IV, E., Kongpensook, V. and Oupadissakoon, C. (2012), Sensory lexicon for mango as affected by cultivars and stages of ripeness. Journal of Sensory Studies, 27: 148-160. https://doi.org/10.1111/j.1745-459X.2012.00377.x
  5. Tran, T., James, M.N., Chambers, D., Koppel, K., and Chambers, IV E. (2019). Lexicon development for the sensory description of rye bread. J Sens Stud. 2019; 34:e12474. https://doi.org/10.1111/joss.12474
  6. Cherdchu, P., Chambers, E., IV and Suwonsichon, T. (2013), Lexicon Development by Two Panels for Soy Sauce. J Sens Stud, 28: 248-255. https://doi.org/10.1111/joss.12041